# Block-Operations: Using Modular Routing to Improve Compositional Generalization

## Abstract

We explore the hypothesis that poor compositional generalization in neural networks is caused by difficulties with learning effective routing. To solve this problem, we propose the concept of block-operations, which is based on splitting all activation tensors in the network into uniformly sized blocks and using an inductive bias to encourage modular routing and modification of these blocks. Based on this concept we introduce the Multiplexer, a new architectural component that enhances the Feed Forward Neural Network (FNN). We experimentally confirm that Multiplexers exhibit strong compositional generalization. On both a synthetic and a realistic task our model was able to learn the underlying process behind the task, whereas both FNNs and Transformers were only able to learn heuristic approximations. We propose as future work to use the principles of block-operations to improve other existing architectures.

## 1 Introduction

Typical artificial neural networks (NNs) perform poorly on tasks of systematic generalization (Marcus, 1998; Bahdanau et al., 2018; Lake & Baroni, 2018). Numerous people have argued that this failure to generalize is caused by Neural Networks' lack of compositionality: The ability to break complex concepts down into their atomic elements and to reuse and recombine them appropriately when faced with a new task (Bahdanau et al., 2018; Lake & Baroni, 2018; Pfeiffer et al., 2023; Barrett et al., 2018; Hupkes et al., 2020; Hill et al., 2019).

In an analysis drawing inspiration from neuroscience and cognitive psychology (Greff et al., 2020) formalized this as the *Symbol Binding Problem*: The problem of representing the world in terms of symbol-like entities and to dynamically bind information that is distributed throughout the network to those symbols. They highlighted that routing in regular neural networks tends to be static, becoming fixed at training time. (Csordás et al., 2020) investigated compositionality in neural networks empirically and found that neural networks tend to learn new feature mappings instead of representation-preserving mappings suitable for routing, even in situations where a consistent representation would clearly be beneficial.

Based on these theoretical analyses and empirical observations, we develop the concept of block-operations in Section 3. Block-operations are a new approach to tackle compositional generalization that is based on building a favorable inductive bias into the network: We first split all network activation tensors into uniformly sized blocks. We then modify network modules to implement Modular Representation-Preserving Mappings (MRPMs), a new method that encourages the network to route and modify blocks dynamically and independently of each other. As we later show, this causes the blocks to act as objects that are used consistently throughout the network and that can be modularly recomposed by changing the order of blocks within a tensor.

We then introduce the Multiplexer and SMFR modules (Stack of Multiplexers and Feedforward Neural Networks with gated residual connections) as a first implementation of block-operations in Section 4. The SMFR implements MRPMs and replaces the basic FNN module.

We ran several experiments to verify the effectiveness of our architecture. In Sections 5.1 and 5.2, we demonstrate useful properties of our module on synthetic data and explore the effects model size and architecture on performance and reliability. In Section 5.3, we show that our module is particularly effective at learning logical rules and variable assignments, such as those seen in algorithmic tasks, beating both the FNN and the Transformer. We show that *the SMFR learned the underlying atomic operations of the task*, which resulted in perfect generalization, whereas the other models only learned statistical correlations. In Section 5.4, we test our module on realistic data: We split MNIST images into four chunks and permute those, creating one new task per permutation. The resulting dataset requires pattern recognition to solve just like MNIST, but can be solved more efficiently if the network understands how to undo the permutations. We found that the SMFR took longer than the FNN or the Transformer to learn this task, but it kept improving in test accuracy and generalization even after training accuracy converged. At longer runtimes, it ended up outperforming the other architectures. Inspecting the models' internals showed that the SMFR correctly learned to decompose the task into its two base tasks: Disentangling the permutation and solving MNIST.

Our contributions are:

- We propose the concept of block-operations, a routing-based approach for making existing neural network architectures better at compositional generalization.

- We apply this concept to create the SMFR module as a replacement for the FNN.

- We empirically demonstrate that SMFRs have the properties we expected and that they outperform both FNNs and Transformers on tasks designed to test compositional generalization.

- We note several remaining limitations and motivate further research on block-operations, which could be used to enhance existing architectures such as Transformers.

## 2 Related Work

**Residual Connections**. Residual connections allow a neural network to route data through the network unchanged, and many variants of this have been used to great success in different areas (Szegedy et al., 2017; He et al., 2016b; Jastrzebski et al., 2017; Bachlechner et al., 2021; Wu et al., 2019; He et al., 2020). The Neural Data Router uses a variant with a learnable interpolation weight and achieved very strong generalization (Csordás et al., 2021). We use this variant for our own architecture.

**Attention**. Attention mechanisms have become a ubiquitous tool in deep learning, useful in a variety of fields such as natural language processing, image processing and speech recognition (Bahdanau et al., 2014; Xu et al., 2015; Vaswani et al., 2017; Devlin et al., 2018; Jaderberg et al., 2015; Chorowski et al., 2015). We note that the term *Attention* is not used consistently in the literature and is often conflated with the more specific *Self-Attention* that is used in Transformers. Attention in Transformers is based on pairwise comparisons. This makes it difficult for them to learn routing mechanisms based on more than two features, such as conditional statements. We experimentally confirm this in Section 5.3. The Multiplexer module we introduce in this paper uses a form of Attention, but not Self-Attention.

**Routing**. Many architectures for routing exist and have been summarized by surveys (Pfeiffer et al., 2023; Han et al., 2021; McGill & Perona, 2017). For a comprehensive overview of related work, we recommend reading (Greff et al., 2020), especially the paragraph on *Instance Slots* on page 13, which is the type of network most similar to our own. *Recurrent Independent Mechanisms* use multiple largely independent cells that communicate through a bottleneck of attention and compete for access to input data (Goyal et al., 2019). The RIM uses multiple modules with their own independent training dynamics, whereas our approach is simpler and uses an inductive bias to allow subnetworks to more easily share data in a consistent format. *Capsule Networks* can perform routing between a fixed number of capsules based on entity recognition (Sabour et al., 2017). They are designed for modeling hierarchical relationships, but have a fundamental design difference to our approach: Each capsule represents a specific type of object, while each block in block-operations is a placeholder variable that can come to hold any type of object. *Slot Attention* (Locatello et al., 2020) comes closer to block-operations as their *slots* are similar to our *blocks*: Each slot can hold a different

object. This architecture is good at extracting object-centric representations, but does not contain a routing mechanism. *Transformers with Competitive Independent Mechanisms (TIM)* split data into blocks, which they call *positions*, and operate on them independently (Lamb et al., 2021). However, they do not include a way to preserve mappings while routing them past a layer of mechanisms, to a subsequent layer of the network. They hypothesize, but do not test, a way to access information from earlier layers, which would be similar to our blockwise routing. However, this is based on Self-Attention and would therefore suffer from the same issues as normal Transformers (see Section 5.3).

## 3  Methodology: Block-Operations

In this section we discuss the Symbol Binding Problem and introduce the concept of block-operations as our approach to solve it.

(Greff et al., 2020) decomposed the binding problem into three different aspects:

- **Representation problem**: Representing multiple objects separately, in a common format, yet without interference between them.

- **Segregation problem**: The ability to form object representations from unstructured inputs.

- **Composition problem**: The ability to dynamically relate and compose these object representations with each other.

Most approaches based on the *instance slots* described by (Greff et al., 2020) rely on weight sharing to encourage consistent representations. Inspired by the problems identified by (Csordás et al., 2020), we take a different approach: *We treat routing as the core of the problem and develop an inductive bias to make learning it as simple as possible.* We hypothesize that if the different components of a neural network can easily learn to share objects with each other, then they will take the path of least resistance and converge towards using a common format even without explicit weight sharing.

However, we also have to take into account the problem of the superposition catastrophe, described by (Von Der Malsburg, 1986): If object representations encoded in a tensor do not have distinguishable roles, it becomes ambiguous which attributes belong to which object. For example, the concepts (red, apple, green, pear), when naively encoded in a single tensor, could either refer to "red apple, green pear" or to "red pear, green apple".

To deal with this problem, block-operations combine two ideas: We split all network activation tensors into uniformly sized blocks, and we require all modules in the network to implement Modular Representation-Preserving Mappings (MRPMs) to operate on these blocks.

**Blocks**. Through the inductive biases provided by MRPMs, each block of a tensor should come to hold a separate object and the position of the block within the tensor should designate its role. If one imagines a neural network module as a function, then the blocks in its input tensor would be a list of discrete function arguments. This discretization mirrors the way humans design algorithms and naturally lends itself to modular decomposition, as demonstrated by our experiments in Section 5.3.

**MRPMs**. It is easiest to think of MRPMs as an enhancement to an existing module that adds optional routing abilities to it. To implement MRPMs, a module must be able to make any of its output blocks a copy of any of its input blocks, and it must be able to decide to do this dynamically, based on its inputs. One way to implement this is to use attention mechanisms or dynamic residual connections to interpolate the module's normal outputs with the input blocks.

MRPMs allow different submodules of the network to easily share data with each other in a representation-preserving manner without hampering their ability to generate new data: The modules can simply use some of their output blocks for their own calculations while using other blocks to pass their inputs through. This easy sharing of data makes it likely that blocks in different parts of the network that represent the same concepts will use the same representations, even if different subnetwork have only encountered different subsets of the concept. Our experiment in Section 5.2 demonstrates this empirically.

**Implementation**. In Section 4 we create the SMFR module as a replacement for the FNN that satisfies the MRPM criteria. It can seamlessly combine two different ways of processing data: Learning new mappings just like a regular FNN, and conditionally routing blocks of data. Figure 1 illustrates the general idea without going into the details. We refer to Section 6 for a discussion on adapting other modules and architectures to block-operations.

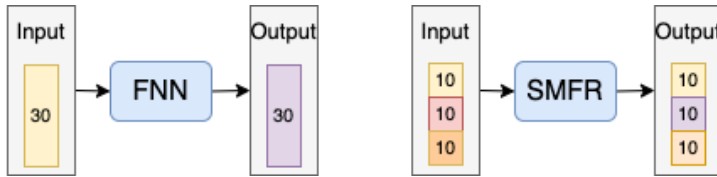

Figure 1: **Left**. An example FNN receives a layer of 30 input neurons and maps it to a layer of 30 output neurons using densely connected layers. **Right.** An equivalent SMFR architecture instead views the input as 3 blocks of 10 neurons each and it outputs another 3 blocks of 10 neurons each. In this example, the first output block is a copy of the first input block. The second output block is generated through an FNN based on all 3 input blocks (the FNN is a submodule inside the SMFR). The third output block is a linear interpolation of the 3 input blocks and an FNN output.

**Tackling the Binding Problem**. Block-operations address all three aspects of the binding problem:

- **Representation problem**: Each block represents one object or feature. Since each feature tensor comprises several blocks, multiple objects can be represented in parallel. The ability to route through MRPMs leads to an inductive bias that encourages all blocks to use the same representation.

- **Segregation problem**: Modules using MRPMs can dynamically decide when to route blocks and when to generate new blocks.

- **Composition problem**: The position of a block within a feature tensor can be changed during routing and influences its semantic meaning: The four-block tensor [red, apple, green, pear] is different from [red, pear, green, apple]. This modularity allows block-operations to avoid the superposition catastrophe.

## 4 Modules

We introduce the **Multiplexer** to route blocks and the **FNNR** (Feedforward Neural Network with gated residuals) to learn new feature mappings. We combine these two modules into the **MFNNR** (Multiplexer plus Feedforward Neural Network with gated residuals), which can do both. Finally, we stack multiple MFNNRs in sequence to form our final architecture, the **SMFR**.

**Multiplexer**. A Multiplexer (Figure 2, left) takes $M$ input blocks and produces $N$ output blocks, each of which is a weighted average of all $M$ input blocks. The weights are generated by an FNN that receives a concatenation of all $M$ blocks as input and outputs an $M * N$ weight matrix, which is normalized by applying a softmax over the first dimension. The Multiplexer can learn to transfer or linearly interpolate blocks. It makes these routing decisions dynamically, for all blocks at the same time, and takes the order of the blocks into account.

**FNNR**. An FNNR (Figure 2, right) takes $N$ input blocks and produces $N$ output blocks. It uses an FNN to generate new blocks and then combines them with the input blocks using residual connections. These residual connections use a learned gating weight instead of simple addition, similar to (Csordás et al., 2021). An FNNR can learn to either let an input block pass through unchanged, or to replace it with a new block created by an internal FNN, or to create a linear interpolation of both. It can make this decision for each of the blocks independently, but can base each decision on all input blocks.

**MFNNR**. An MFNNR is composed of a Multiplexer followed by an FNNR module. The FNNR module uses the input blocks of the Multiplexer as its extra input. The MFNNR can learn to rearrange blocks, or to

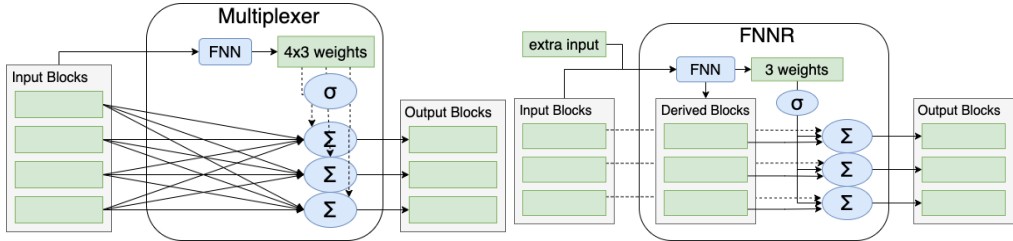

Figure 2: **Left:** A Multiplexer with $M = 4$ and $N = 3$. **Right**: An FNNR with $N = 3$.

generate new blocks through an FNN, or to interpolate between both. It can learn to do this conditionally on the input and with different rules for each output block. An MFNNR can emulate an FNN or a data copying mechanism by setting its gating weights to extreme values. It can learn arbitrary mappings like an FNN, but it also has an inductive bias to copy or interpolate blocks of data if doing so leads to a simpler solution.

**SMFR**. Multiple MFNNR modules can be stacked one after the other to form a more powerful architecture. *The SMFR is the architecture we use in our experiments, as an alternative to FNNs.*

## 5 Experiments

Compositional generalization has multiple different aspects, making it difficult to pick a benchmark for it. A meta analysis comparing different benchmarks on compositional generalization found little concurrence between them (Sun et al., 2023). We therefore chose a combination of specific synthetic tasks, to test the theory, and practical tasks, to confirm our findings against realistic data and strong architectures. We used grid-search to generate different architectures to compare (see appendix).

The first two of our experiments are based on the experiments used by (Csordás et al., 2020) to test a network's ability to specialize and to reuse subnetworks. As the SMFR is intended as a replacement for the FNN, we compare SMFRs with FNNs with a similar numbers of parameters. The experiment in Section 5.1 measures negative interference, i.e. the extent to which altering the training distribution causes the network to forget previously gained knowledge (McCloskey & Cohen, 1989). It shows that *SMFRs are less prone to negative interference than FNNs.* The experiment in Section 5.2 measures generalization through the reuse of subnetworks. It shows that *SMFRs can sometimes learn to reuse subnetworks perfectly even in the absence of training data*, a case of transfer learning.

The experiment in Section 5.3 measures performance on an algorithmic task that requires conditional logic and variable assignments. Here we also perform an ablation study by comparing it to a Transformer that operates on blocks, to demonstrate that existing Attention mechanisms can not efficiently emulate our architecture. Our results show that *SMFRs developed a more accurate model of the underlying atomic operations of the task.*

The experiment in Section 5.4 tests the SMFR's performance on realistic data. We introduce block-permuted MNIST (BPMNIST), a task that can be decomposed into two steps: Learning a subtask-dependent permutation, and solving MNIST. We find that the SMFR took longer to train than the FNN or the Transformer, but its *test accuracy kept increasing even after training accuracy converged*, similar to the "grokking" observed by (Power et al., 2022). Unlike the other architectures, *the SMFR was able to learn the underlying compositional rule* of BPMNIST, allowing it to achieve much better Out-of-Distribution (OOD) accuracy when trained for long enough.

### 5.1 Addition/Multiplication Experiments

**Task**. The addition/multiplication dataset is designed to *test the resilience to negative interference* of a neural network. The task is to either add or multiply two single-digit numbers (modulo 10). We use two training stages. During the preparation stage the multiplication task is trained on limited data and the

addition task is trained on all data. In the limited dataset, one number uses only low digits (0 to 4) and the other number uses only high digits (5 to 9). Once the network reaches a threshold of accuracy, we switch to the negative interference stage, in which the rule for training is inverted: The addition task is trained on limited data and the multiplication task is trained on all data. We run the negative interference stage for a fixed number of additional training steps, then measure the preparation-data accuracy: The accuracy on the data that was only used for training in the preparation stage but not the negative interference stage.

**Variant**. We note that both the addition and the multiplication task are commutative. Our SMFR architecture is good at solving commutative tasks because the softmax used by the Multiplexer is also commutative, giving it a useful inductive bias. This is an additional strength of our architecture over FNNs. We perform an ablation study to measure how much of the SMFR's strength comes from its block-routing abilities and how much from its ability to easily handle commutativity. To do so, we run variants of the experiment in which we use Straight Through Gumbel-Softmax instead of softmax for the Multiplexer, because Gumbel-Softmax makes a discrete pick among candidates and therefore does not help with commutative tasks in the same way softmax does (Jang et al., 2016).

Table 1: Preparation-data accuracy for different thresholds and architectures

| Threshold | FNN | $SMFR_{Softmax}$ | $SMFR_{Gumbel}$ | $SMFR_{Softmax}/FNN$ |
|---|---|---|---|---|
| 0.7 | $0.032 \pm 0.0034$ | $\mathbf{0.142} \pm 0.0101$ | $0.088 \pm 0.0080$ | **4.44** |
| 0.8 | $0.056 \pm 0.0042$ | $\mathbf{0.184} \pm 0.0112$ | $0.119 \pm 0.0099$ | **3.29** |
| 0.9 | $0.123 \pm 0.0068$ | $\mathbf{0.259} \pm 0.0137$ | $0.208 \pm 0.0113$ | **2.11** |
| 0.95 | $0.202 \pm 0.0105$ | $\mathbf{0.326} \pm 0.0143$ | $0.292 \pm 0.0158$ | **1.61** |
| 1.0 | $0.350 \pm 0.0157$ | $\mathbf{0.434} \pm 0.0179$ | $0.395 \pm 0.0182$ | **1.24** |

**Results**. Table 1 shows the preparation-data accuracy of different models. Each line compares averages and standard errors of 90 trials of FNN and 63 different architectures of SMFR for each of softmax and Gumbel-Softmax. The Threshold refers to the accuracy at which we switch to the second stage of training. The values show the preparation-data accuracy 2000 steps after switching to the negative interference stage, except for the last column, which shows the ratio between the values of $SMFR_{Softmax}$ and FNN. We see that $SMFR_{Softmax}$ performs best in all cases and the difference to FNNs is larger for smaller thresholds. In other words, *SMFRs suffer less negative interference than FNNs, especially if the switch between the two training regimes happens earlier.*

**Routing and Commutativity**. $SMFR_{Softmax}$ consistently performs better than $SMFR_{Gumbel}$, and both outperform FNNs. This shows that both the block-routing abilities of SMFRs and their ability to learn commutativity efficiently have positive effects.

**Model Size**. The above analysis is based on an average over all model sizes for both SMFRs and FNNs. An ablation study showed that SMFRs perform better at smaller model sizes and FNNs at larger ones. The SMFRs were still better than the FNNs at larger model sizes, but the difference was less pronounced. The only cases where FNNs slightly exceeded the performance of SMFRs were when the number of parameters was high and we also waited until convergence ($threshold = 1.0$) before starting negative interference. See the appendix for details on this and on architecture optimization.

## 5.2 Double-Addition Experiments

**Task**. The double-addition dataset is designed to *test the ability of a neural network to reuse subnetworks* for similar tasks. The network receives two pairs of numbers and has to return either the sum of the first pair or the sum of the second pair (modulo 10). We want to measure how well the network learns that both tasks require the same logic. To do so, we use a biased training procedure: The distribution of the input numbers is fully uniform for the first pair of numbers, but it is restricted for the second pair of numbers. This uses the same distributions that we also used in the addition/multiplication experiments: One number uses only low digits (0 to 4) and the other number uses only high digits (5 to 9). The training data for the second task is therefore a strict subset of the training data for the first task. During testing, we measure the OOD accuracy on the second pair of numbers.

**Variants**. As before, we run two variants of the SMFRs: One using softmax and one using Straight Through Gumbel-Softmax. Using Gumbel resulted in worse performance overall but otherwise showed the same patterns.

**Results**. FNNs of all model sizes consistently get an OOD accuracy of 0.0, which is actually worse than guessing (0.1). This is because $f(a, b) = a + b$ becomes a bijective function if either $a$ or $b$ are frozen, mapping ten possible inputs to ten possible outputs. The set of possible outputs of the limited training inputs has no overlap with the set of possible outputs of the OOD inputs, which results in an OOD accuracy of 0.0. In contrast, SMFRs reach an average of 0.205 OOD accuracy across all architectures we tested. The surprising part here was that *some trials achieved 100% OOD accuracy*, implying perfect reuse of subnetworks.

**Architecture**. We investigated the effect of the architecture on performance. We found that model size correlated with performance, but bigger was not always better (see the appendix for details). Across all of our experiments with SMFRs, 10.4% achieved 100% OOD accuracy (25 of 240 trials). Best results were achieved when using softmax instead of Gumbel, at a stack depth of exactly 1 and a width of 8 or higher: For these architectures *75% of trials achieved 100% OOD accuracy* (9 of 12). We hypothesize that higher depths make routing more complicated, causing the SMFR to resort back to using FNN logic.

### 5.3 Algorithmic Experiments

**Task**. The ALGO task is designed to emulate patterns of information processing that typically occur in real-life coding tasks. It tests how good the network is at *understanding conditional logic and variable-assignment operations*. The task uses five variables as the input and expects the same five variables as the output. The task proceeds in several iterations, modifying one variable per iteration. On each iteration, a formula of the following form is applied:

"Variables $A, B, C, D$ should remain unaltered. Variable $E$ should be assigned $A + 1$ if $C > D$ and $B + 1$ otherwise."

We use five different permutations of the variables $A, B, C, D$ and $E$. We want to find out if the network learns the actual underlying algorithm, or only a statistical correlate. To do so, we use a special way to train and test the network: During training, we always perform exactly two applications of the rule. As a result, we can test the network for generalization in two different ways: $OOD_{even}$ measures the average accuracy when the network is run for 4, 6, or 8 iterations. This is a form of length generalization. In contrast, $OOD_{odd}$ measures average accuracy for 1, 3, 5, 7, or 9 iterations. Since training happened on 2 iterations, this tests if the network understands that the data generating process can be decomposed into two applications of the same atomic rule.

**Baselines**. In addition to the FNN we also compare to a Transformer. The Transformer operates on the blocks, just like the SMFR, treating the variables on each iteration as a sequence of inputs.

Table 2: The average $OOD_{even}$ and $OOD_{odd}$ accuracies for the five best models of each type. The accuracy at four iterations was used as the validation criterion.

| Architecture | $OOD_{even}$ | $OOD_{odd}$ |
|---|---|---|
| FNN | $\mathbf{1.000} \pm 0.000$ | $0.187 \pm 0.052$ |
| Transformer | $0.984 \pm 0.003$ | $0.099 \pm 0.094$ |
| SMFR | $\mathbf{1.000} \pm 0.000$ | $\mathbf{1.000} \pm 0.000$ |

**Results**. Table 2 summarizes the results. $OOD_{even}$ is high for all architectures, so they all demonstrate good length generalization on this task. The Transformer performed worse than either the FNN or the SMFR. We believe this is because Transformers have an inductive bias that is unsuitable for this task: Applying the task's formula requires considering the values of all variables at the same time, but the Key/Query Attention mechanism in Transformers can only perform pairwise comparisons.

While performance on length generalization was quite similar between architectures, $OOD_{odd}$ was low for both FNNs and Transformers, but the best SMFR architectures actually achieved perfect scores on it. This

indicates that *the SMFR learned the actual underlying rule behind the task*, while the other architectures only learned a heuristic.

**Architecture**. Table 2 only shows the five best-performing models of each type, but the ability of the SMFR to generalize perfectly extended beyond that. For a stack depth between 1 and 5 (we tested 0 to 10) all SMFR models *converged with 100% accuracy on all iterations* regardless of the values of other hyperparameters. This supports our previous finding that the performance of our architecture can be improved by tuning the number of layers. In contrast, both FNNs and Transformers generalized better to greater lengths the larger the model was, but reached their best performance on $OOD_{odd}$ at smaller model sizes, which is a sign of overfitting. We conjecture that larger FNNs and Transformers take the easy way out, so to say, by using their many parameters to just learn extensive statistical mappings. These generalize well to longer sequences but do not capture the actual formula that generates the data. See the appendix for details.

**Working with Noisy Input**. In the above experiments, the inputs were all formatted in a way that is easy for the SMFR to work with because the data is already split into blocks in the appropriate way: Each block represents one variable. This raises the question: What happens if the input is noisy? Can the SMFR manage to reconstruct the correct format? To test this we ran a second set of experiments with a modification: A random but fixed permutation is applied to the state before each iteration, permuting all neurons across all blocks. This random permutation had no effect on FNNs, which still give the same performance since they do not care about the order of neurons in a layer. SMFRs lost some $OOD_{odd}$ accuracy but remained better than FNNs on average. Notably, in one of the 30 trials we ran the SMFR did achieve 100% OOD-accuracy again, even though the inputs were permuted. This shows that *SMFRs can learn to automatically arrange unstructured data into the block format they need*. They can sometimes even do so perfectly, but not reliably. Investigating how to do this more reliably remains as future work.

## 5.4 BPMNIST Experiments

**Task**. The Block-Permuted MNIST task is designed to *test the ability to generalize compositionally on realistic data*. It is based on the popular MNIST task (Deng, 2012). We take the MNIST images and split them horizontally into four equal-sized blocks, then permute these four blocks. [1] The model has to learn to recognize the number for each of eight different permutations. Some numbers are omitted from some permutations in the training set. Therefore, good performance on the test set requires compositional generalization. See the appendix for details on this and the validation set. Intuitively, there are two different ways to solve the task: Learn the mapping for each permutation separately, or learn to reverse the permutation first and then just learn MNIST.

We compared FNNs, Transformers and SMFRs. We noticed that some of the SMFR models continued to strongly improve in test accuracy even after their training accuracy converged, which is similar to the "grokking" behavior observed by (Power et al., 2022). To investigate this, we measured performance at two different time points: After 25k training steps, when the graphs showed that training accuracy had converged on all models, and again after 250k steps, to investigate if any models "grokked" the data.

We also wanted to see if this grokking behavior could be induced artificially. To this end we developed two additional variants of the SMFR. $SMFR_{bias}$ was equipped with an artificial bias for the first 300 training steps only, causing it to learn the ideal permutation quickly. $SMFR_{no\_context}$ used a modified FNNR in which the original input blocks before the Multiplexer are not used as extra input. These inputs are a distraction, because an optimal solution does not need this data to solve this task.

**Results**. Table 3 shows the results. The SMFR models performed worse than the FNNs and the Transformers at lower training times, but they *kept improving after they achieved 100% training accuracy, and eventually surpassed the other models*. The variants $SMFR_{bias}$ and $SMFR_{no\_context}$ both converged much more quickly and the former achieved the best performance of all models. Their strong performance suggests that it could be well worth looking for a way to make SMFRs converge to the optimal routing more quickly. We leave this as future work.

---

[1] Note that this task is different from the "Permuted MNIST" task used by (Goodfellow et al., 2013): We apply the permutation to parts of the image rather than to each individual pixel.

Table 3: The test accuracies, averaged over the five best models of that type according to the validation set.

| Architecture | 25k steps | 250k steps |
|---|---|---|
| FNN | $0.949 \pm 0.00166$ | $0.950 \pm 0.00032$ |
| Transformer | $\mathbf{0.975} \pm 0.00034$ | $0.977 \pm 0.00083$ |
| SMFR | $0.945 \pm 0.00980$ | $\mathbf{0.985} \pm 0.00092$ |
| $\text{SMFR}_{bias}$ | $\mathbf{0.984} \pm 0.00045$ | $\mathbf{0.986} \pm 0.00111$ |
| $\text{SMFR}_{no\_context}$ | $0.982 \pm 0.00060$ | $0.984 \pm 0.00267$ |

**Analysis**. For SMFRs and Transformers, we recorded key indicators of internal network behavior to try to understand how they learned to solve the problem. The SMFRs that performed well tended to rearrange the four partial images in the very first MFNNR layer to get the same mapping for each permutation. The longer training progressed, the more of the SMFRs converged to this behavior. Unlike the SMFRs, Transformers performed extremely well early on, but then barely improved with longer runtime. It appears that they simply learned to take an average over all input blocks, which is the same for all permutations, and then learned to detect MNIST digits from this mix. This heuristic is easy to learn and worked surprisingly well, but did not leave room for further improvement. The Transformer models ended up stuck in this local optimum.

**Effects of Contained FNN Size**. The SMFR contains FNNs, but since much of the model's parameter count is used for routing operations, the contained FNN is necessarily smaller than the pure FNNs we used in our comparison. We ran some additional tests to find out how important the size of these contained networks is relative to the ability to perform routing. To do so, we took the best SMFR architecture and increased the size of its contained FNNs to be equal to the standalone FNNs. The resulting models achieved an accuracy of 0.948 after 25k training steps. This was better than before, but still slightly worse than the FNN. It appears that until the network figures out how to route effectively, the overhead of doing so can sometimes do more harm than good.

## 6 Discussion

**Block-Operations in Other Architectures**. Adjusting other architectures to block-operations requires both the use of blocks and of MRPMs throughout the entire architecture. Any module that does not support MRPMs will act as a barrier that prevents efficient communication. These modifications are non-trivial and we leave them as future work. Additionally, since FNNs are so fundamental, it is possible that replacing all of them inside large, established architectures will have harmful side effects that require additional research to fix. Nevertheless, since the SMFR is both useful in its current state and shows room for improvement through variants and architecture optimization, we believe that experimenting with different implementations of block-operations is a promising line of research to improve compositional generalization.

Transformers are a good candidate for enhancement. Several previous papers have improved a Transformer's ability to generalize compositionally by providing ways to route input around them (Csordás et al., 2021; Ontañón et al., 2021). The similar but more thorough modifications of MRPMs will likely bring similar benefits.

Since block-operations focus on *routing* representations we also expect that they will go well together with existing architectures that focus on *generating* object representations, such as Capsule Networks (Sabour et al., 2017), Slot Attention mechanisms (Locatello et al., 2020), or TIMs (Lamb et al., 2021).

**Commutativity and Argument Selection**. Our module exhibits an inductive bias that helps it to learn tasks that rely on commutative functions, or on selecting arguments from a set of options. Commutativity and argument selection are fundamental concepts in mathematics and programming respectively, so learning them more easily is likely to be helpful for these types of tasks.

**Interpretability**. In the course of our experiments we noticed that SMFRs were easier to interpret than FNNs. The activations of the softmax and sigmoid neurons that control the routing correlate with the use

of different subtasks. This is analogous to how Transformers can be inspected by visually highlighting how much attention the model pays to different words (Tenney et al., 2019). See the appendix for details.

## 7 Limitations

**Architecture Optimization**. Our experiments showed that increasing an SMFR's depth and width does not always improve performance. FNNs used to have the same issue, until (He et al., 2016a) introduced residual connections. We hope that a similar fix can be discovered for SMFRs as well.

**Computational Overhead**. SMFRs take more time per training step than FNNs of equal size because they perform a large number of operations on small matrices. We have found empirically that they take more time than FNNs to converge on tasks that require heuristic reasoning but less on tasks with a lot of compositional logic, where their useful inductive bias outweighs the computational overhead.

**Stability**. We have found that the SMFR architecture can sometimes get stuck in local optima because the gating weights and softmax values take on too extreme values, which kills the gradient. We fixed this by adding a regularization loss: Whenever the absolute of the weight used for a softmax or sigmoid exceeds a threshold value, we apply an MSE-loss to that weight, pushing it back to the threshold. See the appendix for details.

## 8 Conclusion

Neural Networks have difficulty achieving compositional generalization because it is difficult for them to route objects between subnetworks without incidentally modifying them. We introduced the idea of block-operations, which is based on aggregating network activation neurons into larger semantic units and giving modules an inductive bias to route or modify blocks independently of each other. Based on this idea, we developed the SMFR as a replacement for the FNN. In our experiments the SMFR demonstrated superior generalization compared to other models and exhibited numerous useful properties, such as reusing sub-networks, understanding algorithmic logic, and being easy to inspect. In several experiments it learned to decompose a task correctly where other architectures failed to do so. This indicates that the SMFR is helpful for tackling the challenging problem of compositional generalization. We noted several remaining limitations and proposed future directions of research. Additionally, our findings suggest that the underlying idea of block-operations may be useful to other researchers in neural architecture design.

### Broader Impact Statement

The research presented in this paper is fundamental research and has no direct societal impact. However, the research presented here is low-probability, high-impact. Implementing block-operations in existing large-scale architectures may fail to be useful at all for some architectures, but cause an explosive and unexpected advance in others. It may also cause models to become better at one area but worse at others, and this shift in capabilities will be difficult to predict in advance. Any sudden and substantial change in the performance of a model can have wide-reaching implications in whatever field the model is deployed.

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

# A    Appendix

## A.1    Code

The supplementary material contains an executable python script that runs a trial of the algorithmic experiment in Section 5.3. It includes pytorch modules for the Multiplexer, the FNNR, the MFNNR, and the SMFR. These modules are written with understandability in mind rather than performance.

An updated version with more details and more efficient modules will be provided in a github repository on acceptance of the paper, as it takes time to extract the relevant parts from our code base.

## A.2 Experiment Details, General

We used grid-search to generate different architectures. For FNNs we varied the number and size of intermediate layers. For SMFRs we varied the number of MFNNR modules (depth) and the number of blocks in each layer of neurons between the MFNNR modules (width). A depth of zero means that only a single MFNNR is used to map the input blocks to the output blocks directly. We also varied the size of the FNNs inside the MFNNR modules, but only to a lesser extent. For Transformers we varied the number of heads, the internal dimensions, and the depth of the encoder and decoder.

For all of our experiments, we used an Adam optimizer with a learning rate of 3e-4 and gradient clipping at 0.1. We used LeakyRelu with a negative slope of 0.01 as the activation function. The FNNs contained inside SMFRs had a width of 100 and variable depth, unless stated otherwise.

Each of the first three experiments used a set of digits as inputs, as well as a task indicator to differentiate between the subtasks of the experiment. The digits were one-hot encoded, with a block size of ten. The task-indicator tensor became its own block and was padded to the block size. The experiment in Section 5.4 used a block size of 196 instead, which is one fourth of the size of an MNIST image.

We used single-digit numbers in the addition/multiplication experiment and the double-addition experiment. We ran some exploratory tests with two digits as well and got similar results. We focused our analysis on single digit experiments because those converged faster, so we could run more trials.

### A.2.1 Experiment Details: Addition/Multiplication Experiments

We performed grid search over the following parameters:

- The threshold value before switching training regimes: 0.7, 0.8, 0.9, 0.95, 1.0

- For FNNs
    - Layer widths: 100, 200, 300
    - Layer depths: 2, 3, 4

- For SMFRs
    - Stack widths: 1, 2, 3, 4, 5, 6, 7
    - Stack depths: 1, 2, 3
    - Contained FNN layer depths: 1, 2, 3
    - Attention type: Softmax, Straight Through Gumbel-Softmax

This resulted in model sizes in the range of roughly 13k to 280k parameters for both FNNs and SMFRs.

Table 4 shows the effect of model size on the preparation-data accuracy after catastrophic interference. HIGH refers to all models with at least 200,000 parameters, LOW refers to models with less than 50,000 parameters, and MID refers to the rest. The table shows that FNNs perform better at higher model sizes. It should be noted that the model sizes of FNNs and SMFRs were varied in different ways. The table indicates that growing SMFRs by stacking more MFNNR modules together is not as effective as increasing the FNNs inside the SMFRs.

**Architecture Optimization**. Note that we varied the model sizes of FNNs and SMFRs in different ways, and this may explain some of our findings. Table 5 shows that SMFRs with more complex FNNs inside them performed better, especially at low thresholds. However this also had diminishing returns at 3 layers. Our tests only included small variations on the contained FNNs because we focused on varying the number of blocks and MFNNR modules instead of the complexity of each FNN contained within them. Additionally, we only altered the depths of the contained FNNs, not their width as well, as we did for the baseline FNNs

Table 4: Effects of model size on performance

| Threshold | Architecture | Softmax/Gumbel | *HIGH* | *MID* | *LOW* |
|---|---|---|---|---|---|
| 0.7 | FNN | | 0.054 | 0.024 | 0.035 |
| 0.7 | SMFR | Softmax | **0.204** | 0.146 | 0.107 |
| 0.7 | SMFR | Gumbel | 0.146 | 0.094 | 0.049 |
| 0.8 | FNN | | 0.089 | 0.053 | 0.052 |
| 0.8 | SMFR | Softmax | 0.186 | **0.205** | 0.116 |
| 0.8 | SMFR | Gumbel | 0.118 | 0.138 | 0.063 |
| 0.9 | FNN | | 0.179 | 0.134 | 0.098 |
| 0.9 | SMFR | Softmax | **0.300** | 0.274 | 0.198 |
| 0.9 | SMFR | Gumbel | 0.266 | 0.232 | 0.109 |
| 0.95 | FNN | | 0.325 | 0.223 | 0.150 |
| 0.95 | SMFR | Softmax | 0.346 | **0.349** | 0.245 |
| 0.95 | SMFR | Gumbel | 0.274 | 0.333 | 0.172 |
| 1.0 | FNN | | **0.468** | 0.413 | 0.259 |
| 1.0 | SMFR | Softmax | 0.450 | 0.448 | 0.382 |
| 1.0 | SMFR | Gumbel | 0.408 | 0.415 | 0.327 |

Table 5: Complexity of FNNs within MFNNRs

| Threshold | 1 hidden layer | 2 hidden layers | 3 hidden layers |
|---|---|---|---|
| 0.7 | 0.099 | 0.134 | **0.192** |
| 0.8 | 0.125 | **0.216** | 0.210 |
| 0.9 | 0.194 | **0.299** | 0.284 |
| 0.95 | 0.275 | **0.365** | 0.338 |
| 1.0 | 0.365 | **0.489** | 0.449 |

that weren't part of an MFNNR. For FNNs, altering the width made a large difference. Increasing the width of the FNNs inside the SMFRs as well would have led to much larger SMFR models, which is why we did not test this. Therefore, the reason that pure FNNs performed slightly better than SMFRs at high model sizes and thresholds may be because we grew the complexity of SMFRs in a suboptimal way. It remains future work to test if tuning the FNNs inside the SMFRs gives even better results than varying the number of blocks and MFNNR modules.

### A.2.2 Experiment Details: Double-Addition Experiments

We performed grid search over the following parameters:

- For FNNs
  - Layer widths: 100, 200, 300
  - Layer depths: 2, 3, 4

- For SMFRs
  - Stack widths: 1, 2, 3, 4, 5, 6, 7, 8, 9, 10
  - Stack depths: 0, 1, 2
  - Contained FNN layer depths: 1, 2
  - Attention type: Softmax, Straight Through Gumbel-Softmax

Module size comparisons between FNN and SMFR are largely irrelevant here because all FNNs failed regardless of size.

Table 6: Best architectures for the double-addition task

| alternate split | Attention type | depth | width | FNN layers | model size |
|---|---|---|---|---|---|
| True | soft | 1 | 9 | 2 | 95364 |
| True | soft | 1 | 5 | 1 | 36096 |
| False | soft | 1 | 9 | 1 | 54964 |
| False | soft | 1 | 8 | 1 | 50247 |
| False | soft | 1 | 10 | 1 | 59681 |
| False | soft | 1 | 7 | 1 | 45530 |
| False | gumbel | 1 | 8 | 1 | 50249 |
| False | soft | 1 | 6 | 1 | 40813 |
| True | soft | 1 | 10 | 1 | 59681 |
| True | soft | 1 | 9 | 1 | 54964 |
| True | soft | 1 | 8 | 1 | 50247 |
| True | soft | 1 | 7 | 1 | 45530 |
| True | soft | 1 | 6 | 1 | 40813 |
| False | gumbel | 1 | 10 | 1 | 59683 |
| True | soft | 2 | 10 | 1 | 111091 |
| True | gumbel | 1 | 6 | 1 | 40815 |
| False | gumbel | 1 | 7 | 1 | 45532 |
| False | gumbel | 1 | 6 | 1 | 40815 |
| True | soft | 1 | 6 | 2 | 81213 |
| True | soft | 1 | 8 | 2 | 90647 |
| True | gumbel | 1 | 9 | 1 | 54966 |
| True | soft | 1 | 10 | 2 | 100081 |
| True | soft | 2 | 9 | 2 | 160944 |
| True | soft | 2 | 5 | 2 | 119976 |
| False | gumbel | 1 | 9 | 1 | 54966 |

Table 6 shows all 25 architectures with an OOD accuracy of 1.0, because a detailed breakdown over several dimensions is much more confusing to look at. This shows that model size correlates with performance. However, bigger models are not always better because the number of FNN layers was often 1 instead of 2, even though this parameter has a large impact on the model size.

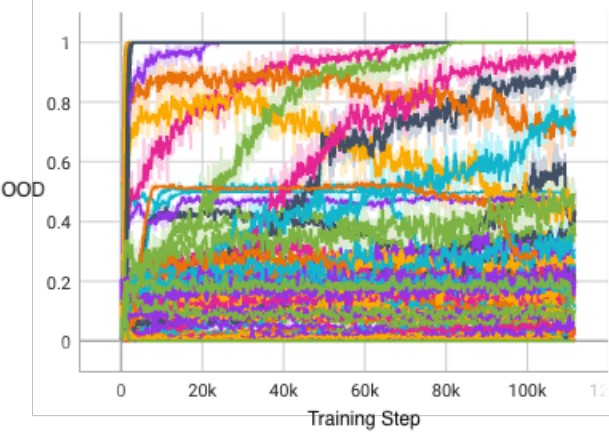

Figure 3: Each line shows the OOD accuracy of one SMFR experiment as training progresses.

**Convergence Behavior**. Figure 3 shows examples of the development of the OOD accuracy over time for different SMFR architectures. Each line represents one training run. The colors are meaningless and are just

there to make it possible to tell the lines apart. Most experiments with a perfect OOD accuracy converged to 1.0 almost immediately. In cases where this didn't happen, convergence took significantly longer, presumably because the model gets stuck in a local optimum and takes a while to escape. It happened more often that OOD accuracy increased than that it dropped. Perhaps most importantly, *the OOD accuracy never dropped after reaching 1.0*. In other words, the SMFR usually came to reuse modules more rather than less as the training progressed and remained stable once converged. This suggests that subnetwork reuse will also occur if SMFRs are used as components inside larger architectures that train for longer.

### A.2.3   Experiment Details: Algorithmic Experiments

We performed grid search over the following parameters:

- For FNNs
    - Layer widths: 100, 200, 300
    - Layer depths: 2, 3, 4, 5, 6

- For Transformers
    - Number of heads: 1, 4, 8
    - Internal dimensions: 400, 1200, 2000
    - Number of Encoder and Decoder Layers: 1, 2, 3, 4, 5, 6

- For SMFRs
    - Stack widths: 6, 8, 10
    - Stack depths: 0, 1, 2, 3, 4, 5, 6, 7, 8, 9, 10
    - Contained FNN layer depths: 1, 2, 3

This resulted in model sizes in the range of roughly 20k to 500k parameters for all three types of architectures.

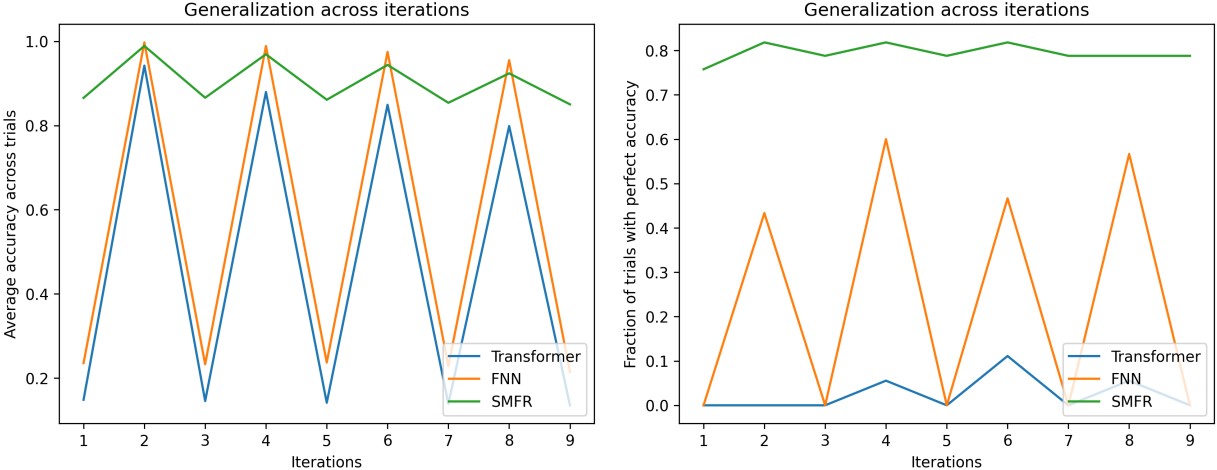

Figure 4: **Left**: The average accuracy of different architectures and variants at different iterations. **Right**: The fraction of trials that achieved 100% accuracy, instead of the average accuracy. The zig-zag pattern is intended behavior: We train on 2 iterations, so odd-numbered iterations are OOD in a different way than even-numbered ones.

Figure 4 shows how the average accuracy of the different architectures changes with the number of iterations. Note that the training accuracy is the entry at *iterations* = 2. Unlike the data reported in the main paper, which was based on the best models only, these graphs show averages over all models we ran, to provide a better overview.

Table 7: Effect of model size on OOD performance, as averages over all models.

| Architecture | model size category | $OOD_{even}$ | $OOD_{odd}$ |
|---|---|---|---|
| FNN | high | **1.000** | 0.241 |
| FNN | mid | 0.989 | **0.249** |
| FNN | low | 0.891 | 0.164 |
| Transformer | high | **0.967** | 0.159 |
| Transformer | mid | 0.801 | 0.081 |
| Transformer | low | 0.675 | **0.393** |
| SMFR | high | 0.954 | 0.900 |
| SMFR | mid | **0.966** | **0.951** |
| SMFR | low | 0.812 | 0.229 |

Table 7 shows the average $OOD_{even}$ and $OOD_{odd}$ accuracies for models of different sizes. As pointed out in the main text, the SMFR has a sweet spot in performance based on the depth of the architecture. In contrast, FNN and Transformer models become better at length generalization ($OOD_{even}$) as the model size increases, but do not become better at understanding the underlying atomic operation of the task ($OOD_{odd}$).

### A.2.4    Experiment Details: BPMNIST

We used eight different permutations for the task. In order to avoid unintentional bias, we selected the permutations in such a way that each block appears in each position an equal number of times when averaged over all permutations. We reported averages over the five best trials. The validation accuracy is the average over the first four permutations, the test accuracy is the average over the last four permutations. For each of the permutations used for testing we removed one of the ten numbers from the training set.

Because larger FNN models always performed better than smaller ones on this task, we did not use grid search as before. Instead, we varied the parameters mentioned in the previous experiment while keeping the total parameter count of each model roughly the same, around 4 million parameters. We also increased the standard size of the FNNs contained within an SMFR to have a depth of 4 and a width of 400, which was larger than before but still smaller than the pure FNN. The best SMFRs had a low stack depth, which also made them easier to inspect.

We inspected the internal network behavior of Transformers and SMFRs by graphing how the following indicators developed over time:

- Attention sharpness: How much does each attention mechanism pay attention to only a single input vs. all of them? Good SMFR runs had high sharpness.

- Difference between permutations: If we apply the inverse of the permutation used in a training batch to the data after the first attention layer, do we get similar distributions for each permutation? This measures roughly how much the network learns to undo the permutation. Good SMFR runs did learn to undo permutations.

- Fairness of attention: Does the network pay equal amounts of attention to each of the four input blocks, or does it drop some of them? MNIST images carry most of their information content in the middle of the image and not the borders, so the outer two blocks were sometimes ignored by the model. The best-performing runs learned not to throw away this data.

### A.3    Regularization Loss

As mentioned in the Limitations, our model used to suffer from a stability issue that we have since fixed. The problem was that softmax and sigmoid weights sometimes kept growing even after becoming extreme enough that they had a gradient of zero. Our inspection showed that this was caused by the fact that the softmax

and sigmoid values are based on computations that are also used for generating data in the MFNNRs. This means that they are subject to destructive interference. They effectively receive gradients from two sources: The softmax/sigmoid they are supposed to work on and incidental gradients from other parts of the model, but the former irrecoverably become zero.

We fixed this by adding a regularization loss: Whenever the absolute of the weight used for a softmax or sigmoid exceeds a threshold value, we apply an MSE-loss to that weight, pushing it back to the threshold. The target value of the MSE-loss is simply the tensor itself with each neuron clamped to a threshold value. We empirically found that a threshold of 20 worked well.

It should be noted that we introduced the regularization loss while working on a larger, more complex experiment that we will report in a subsequent paper. It is possible that this loss is not needed for many of the experiments we report in this paper. We have noticed that using it slows convergence but increases reliability. We have opted to always use it because reliability is more important to us than convergence speed while developing new architectures.

## A.4 Interpretability

To inspect an SMFR model, it is often enough to look at the weights that determine routing decisions: The softmax values in the Multiplexer, and the gating weight in the FNNR. It can be helpful to have a set of example inputs for which you expect different routing behavior and to compare the routing weights for these.

On the double-addition task, our inspection showed that *it is possible to measure the degree of modularity in our SMFR directly by investigating the softmax weights on example inputs*. We manually inspected the values of the weights that the multiplexers and FNNR modules use in a trial that achieved good OOD performance and had a simple architecture, with depth 1 and width 2. The architecture had only 12 Multiplexer weights and 3 FNNR residual weights, which made it very easy to investigate how data is routed in the network for any given input. Our inspection showed that the irrelevant inputs did correctly receive a weight of 0, while the ones that are important received a weight of 0.5. As it turns out, the network learned to use the correct pair of numbers and used interpolation in a single block to take advantage of symmetry. Instead of always moving block $A1$ to $B1$ and $A2$ to $B2$ as a human would intuitively do it, it learned to interpolate $B1 = 0.5 * A1 + 0.5 * A2$, which is actually more efficient. We also found that the gating weights of several FNNR modules approached zero as training progressed. This implied that the network came to rely more and more on copy operations and less on FNN operations. It was quite easy to inspect the network and learn this information, since we only needed to inspect the gating weights. In larger networks used for practical tasks, the number of network weights may be too large for a manual analysis. However there should still be few enough of them for an automated analysis to yield useful results, especially since the gating weights have inherent meaning (does a block of data get used or not?), unlike normal neurons in a fully-connected neural network, which can mean anything.

On the algorithmic task, our inspection confirmed that the SMFRs can learn to pass blocks through without altering them where doing so is useful, and to alter only the ones that need to be altered. We were able to confirm this desired behavior by only looking at the routing weights. In an FNN, confirming such behavior would have required a much more complex correlation analysis of the neurons.

On the BPMNIST task, our inspection allowed us to determine which models understood the data generating process by decomposing the task into its two subtasks (fixing the permutation and solving MNIST). We reported on this in detail in Section 5.4 and Section A.2.4 of the appendix.

