# OpenReview forum: "Block-Operations: Using Modular Routing to Improve Compositional Generalization"
_TMLR — Rejected by TMLR_

### Review · Reviewer_8qAF · 2025-02-01

**Summary Of Contributions:**

The paper hypothesizes that the poor compositional generalization seen with neural nets is due to difficulties inherent in existing architectures with learning effective routing. The paper proposes  a new architecture called Stack of Multiplexers and Feedforward nets with gated Residual connections (SMFR) as a solution. The paper compares the performance of SMFR with that of MLPs (FFNs) and Transformers with synthetic and small scale datasets. Experimental results are provide that suggest SMFR performs better than baselines on datasets considered in the experiments.

**Audience:**

No

**Broader Impact Concerns:**

Paper has a broad impact statement which is good

**Claims And Evidence:**

No

**Requested Changes:**

-  Q1: A large scale experiment that illustrates the scaling ability of SFMR would be very convincing to readers. Would it be possible to include this type of experiment in the paper?

- Q2:  Would it be possible to provide results with datasets like CLEVR (and others that are used in compositional generalization subfield). This question is related to Q1

- Q3: The paper briefly mentions interpretability but does not go deep into this aspect of the architecture analysis. Please expand on this aspect

- Q4: If possible, please consider the gating mechanism used in MoEs as a baseline. If not, a discussion on why this might not be the right choice would be good to have in the paper

**Strengths And Weaknesses:**

# Strengths

- S1: The paper studies an interesting and important problem of compositional generalization. This problem is of broad interest to the community as  they may be a building block used to builds systems that may have the the ability to reason

- S2: The paper provides a clear explanation of why dynamic routing may play an important role for compositional generalization

- S3: The paper provides an implementation that is built up from first principles with a reasonably clear explanation (note that it took a couple of readings for this reviewer to properly comprehend the implementation but I cannot think of anything to suggest that can improve writing so leaving this observation as a strength)

# Weaknesses

- W1 The architecture described in the paper reminds the reviewer of the feed forward nets (FFNs) used in mixture of experts (MoEs). The gating mechanism used in MoEs could perhaps be an interesting choice as a baseline over Gumberl-Softmax. Have the authors considered this architecture in their work?

- W2: The datasets used in the paper are small-scale and appear to be a good "probe" rather than datasets that can convince the general audience on the merits of the architecture.

- W3: The paper states that achieving perfect accuracy suggests that network learns the underlying rule in Section 5.3 on length generalization. This point is not clear to the reviewer. Why is this statement true? Is there a way to guarantee that the network is not capturing statistical patterns but the actual underlying rule.

- W4: The MNIST-based example while neat appears to be too simple. Would the network perform similarly if one were to use Fashion MNIST instead of MNIST?

- W5: The paper mentions interpretability but does not conduct a convincing analysis to show how the optimized networks are interpretable

- W6: The writing of experiments section in the paper requires more work. The tables provided to support experiments are hard to comprehend due to missing scale of numbers in Section 5.1 (accuracy is from 0 - 1 while one could reasonably expect 0 %- 100 %). Section 5.2 describes numbers while it may be preferable to have these results in a table

Overall, the paper as written currently has a focus that's very narrow and may not have convincing evidence to support its claims and also be interesting to a subset of researchers in TMLR (subjective). I would like to hear the authors rebuttal to either correct my misunderstanding or see the paper be improved for this reviewer to support acceptance

---

> ### Author Response · Authors · 2025-02-03
>
> We thank the reviewer for their detailed feedback and address each point below:
>
> W1: There are similarities between our model and MOEs in terms of the algorithms used for implementation. However, the softmax variant of our architecture is actually closer to MOEs than the gumbel-softmax and while the formulas are similar, they are used in different ways and have a different effect: The SMFR uses the softmax for recombining objects rather than for prioritizing between data and/or experts. Our method could be combined with MOEs to ensure that the experts can communicate more effectively with each other. As an analogy: MOEs instantiate experts and allow them to communicate. SMFRs would be employed *within* the experts and this would allow the agents to communicate with each other more effectively, by making it easier for them to unify the representation of the data in messages between them. We will expand the Related Work section to explain details and add an illustration of the difference.
>
> W2: The datasets were chosen to be minimal model organisms that focus on proving the traits we wanted to prove, while avoiding confounders. We chose to use smaller datasets instead of larger ones so that we could run many trials and verify through the low variance between results that our experiments are reproducible and not the result of luck.
>
> W3: We investigated the values of the gates in Appendix 4. Each gate used exactly the data sources that are relevant to produce the underlying rule, and no others. Theoretically this could be coincidence, but this seems very unlikely to us. Once the network is already targeting exactly the correct information sources, it no longer has a need to rely on statistics because the exact relationship is now trivial to learn.
>
> W4: We see no reason why it would not work on Fashion MNIST since the two problems are almost identical in what they test, just different in scale. We chose a smaller dataset to be able to run more experiments.
>
> W5: We provide details on interpretability in Appendix 4. We will be happy to move it to the main section and add more details.
>
> W6: We will adjust this to increase readability
>
> Q1: As we discuss in Section 6, adjusting other architectures than pure FNNs will require additional theoretical work. While we fully intend to do this, the SMFR is already quite complicated as it is, as you mention yourself under S3. We could run larger datasets on larger FNNs, but we expect that the results of those experiments would not be qualitatively different from our existing experiments. Frankly, we have to choose between “risk getting the paper rejected because there is no comparison to SOTA Transformer architectures” and “definitely get rejected because the paper introduces too many new architectures at the same time and is incomprehensible and over the page limit”. In our opinion, this paper has sufficient novelty and sufficient proof that it can stand on its own. Papers building on the concept of block-operations and adjusting other architectures to them could then build on this paper.
>
> Q2: Working with CLEVR requires a neural network that can work with images. We would be happy to do this in follow-up work: We have ideas for how to adjust CNNs to be compatible with block-operations, but this will require introducing another new architecture.
>
> Q3: See our response to W5.
>
> Q4: See our response to W1.

---

> > ### Comment · Reviewer_8qAF · 2025-02-20
> > **Thanks for the rebuttal**
> >
> > THanks to the authors for their rebuttal. I have read the rebuttal and noted my responses and acknowledgments below
> >
> > > W1: There are similarities between our model and MOEs in terms of the algorithms used for implementation. However, the softmax variant of our architecture is actually closer to MOEs than the gumbel-softmax and while the formulas are similar, they are used in different ways and have a different effect: The SMFR uses the softmax for recombining objects rather than for prioritizing between data and/or experts. Our method could be combined with MOEs to ensure that the experts can communicate more effectively with each other. As an analogy: MOEs instantiate experts and allow them to communicate. SMFRs would be employed within the experts and this would allow the agents to communicate with each other more effectively, by making it easier for them to unify the representation of the data in messages between them. We will expand the Related Work section to explain details and add an illustration of the difference.
> >
> > A note in the related works section would be great to see (but I leave the decision to include/exclude to the authors discretion)
> >
> > > W2: The datasets were chosen to be minimal model organisms that focus on proving the traits we wanted to prove, while avoiding confounders. We chose to use smaller datasets instead of larger ones so that we could run many trials and verify through the low variance between results that our experiments are reproducible and not the result of luck.
> >
> > In general, I agree with the authors approach to minimal model organisms. My concern is whether the datasets are sufficiently large, useful and interesting enough for the community to build on SFMR. Given the empirical nature of the paper,  a reader might reasonably ask whether the findings will translate to datasets that are closer to "real datasets".
> >
> > > W3
> > OK noted
> >
> > > W4: We see no reason why it would not work on Fashion MNIST since the two problems are almost identical in what they test, just different in scale. We chose a smaller dataset to be able to run more experiments.
> > OK. I am not sure what is meant by smaller above but Fashion MNIST is identical in terms of sample count etc to original MNIST. A more complicated dataset would be CIFAR-10 which is similar in scale to MNIST but has "natural content".
> >
> > > Authors response to W5 and W6
> > Yes please make the changes in your next revision.
> >
> > > Authors response to Q1
> > Noted. Thank you.
> >
> > It would be helpful if the changes that authors commit to are made available to the reviewers prior to discussion with AE. I have no other questions at the moment.

---

### Review · Reviewer_U78F · 2025-02-03

**Summary Of Contributions:**

This work attempts to address limitations of compositionality in neural networks using an approach that encourages specific information routing operations. This approach contrasts with prior work that focuses on instilling an inductive bias (using weight sharing) towards sharing a common representational format that can be operated on in a uniform manner. The authors provide a description of their SMFR module, which implements such a routing inductive bias, and test it on several algorithmic tasks, some of which have been previously studied. This module is benchmarked against standard feedforward networks and transformers.

**Audience:**

No

**Claims And Evidence:**

No

**Requested Changes:**

Major Changes

Better describe a path towards scaling this method to larger sequence lengths.

Test the proposed method on at least one existing compositionality benchmark.

Minor Changes: See above, under “Weaknesses”

**Strengths And Weaknesses:**

Strengths: The proposed approach is well-motivated, providing a reasonable approach to addressing the symbol binding problem in simple settings. Though fairly complex, the various components of the SMFR module directly map on to the three aspects of the binding problem.  The authors provide a compelling reason (i.e. pairwise comparisons in self-attention) to doubt the suitability of transformers for such routing tasks. Several of the experiments are interesting and illustrate the strengths of the module.

Weaknesses: The greatest weakness of the submission is one that the authors themselves note: it is unclear how this module might actually be integrated into any existing architecture, or how it might scale up to solve any sufficiently complex problem. Section 7 mentions issues of stability, computational overhead, and architecture optimization, but another fundamental issue is the sheer number of parameters required for each of these blocks. Whereas a transformer shares MLP weights for each block at each token position, this approach requires not one but two FFNs that operate across the whole sequence. It is thus very unclear how this might scale to nontrivial sequence lengths, or whether the method can even handle varying sequence lengths. These are core issues for nearly every interesting domain. The experiment described in section 5.3 is described as an instance of length generalization, but it is (as I understand it) more accurately described as an instance of computational depth generalization, and thus does not address this problem.

Perhaps relatedly, the second major point of concern is that very few of the existing compositional reasoning benchmarks are employed in this work. Though the arithmetic and algorithmic experiments are great case studies for describing the benefits of the method, the “real-world” task is quite contrived. The authors note that plenty of compositional reasoning benchmarks exist in the literature (CLVR, CVR [https://arxiv.org/abs/2206.05379], COGS, SCAN), yet choose to add another that is tailor made to suit their method. This work would be greatly improved by putting their method in conversation with existing benchmarks.

Beyond these main concerns, there are several more minor weaknesses that the authors might address.

1. When describing performance vs. model size, it would be much stronger and clearer to simply include a graph with total parameter count (not layer count, width, etc) vs. performance for the SMFR variants and baseline models. It is quite difficult to understand how the different architectural hyperparameters map onto total parameter count, and so this makes direct comparison tricky. The binning thresholds that define High/Mid/Low (as is done in Table 4 and 7) seem to be somewhat arbitrary, and lead one to wonder how the results shake out when they are disaggregated.

2. For section 5.1: This surely works if you use limited data for addition during the preparation stage and then limited data for multiplication during the negative interference stage, right? If so, why not run this symmetric version as well and average the results? Omitting this experiment leaves the reader wondering whether they are missing something.

3. In Section 5.1, more time should be devoted to explaining the Gumbel-Softmax variant. Readers unfamiliar with this technique will find this very confusing.

4. In general, some of the interpretability experiments might be moved to the main body, or otherwise more clearly referred to in the main text. The description of the interpretability work is somewhat hand-wavey, and could benefit from a more formal treatment and/or supporting figures.

5. Section 5.3: What is the supervision signal here?

---

> ### Author Response · Authors · 2025-02-04
>
> We thank the reviewer for their detailed feedback and address each point below:
>
> On integration: It is not unclear how to integrate the architecture, it just involves several steps and so we have omitted these for brevity and because they were not directly relevant to the experiments. We have derived versions of many existing architectural building blocks that are compatible with block-operations. We can include them if requested.
>
> On the number of parameters: While there are several FNNs, the one in the SMFR can be much smaller than the one in the FNNR.
>
> On length generalization: The SMFR is a replacement of FNNs and is not meant to handle length generalization on its own. The concept of block-operations could also be applied to modify Transformers or LSTMS. The resulting hybrid architecture can then combine the sequence-processing abilities of transformers/LSTMS with the compositional generalization abilities of block-operations.
>
> On the benchmarks: “Compositionality” is unfortunately an extremely loaded term in the literature. As we mention in Section 5, a meta-analysis showed that each of these benchmarks tests for different things, as there is little correlation between models and their performances on these benchmarks. People keep trying and failing to capture an intuitive notion of generalization with semi-realistic data. To avoid making the same mistake, we decided to go in the other direction for the first three experiments: The experiments were simple, but tested for specific and very broadly applicable properties: negative interference, subnetwork reuse, and conditional logic. To touch on your figure of speech: You mention putting our method in conversation with existing benchmarks, but those benchmarks are not talking to each other in the first place. As shown by https://arxiv.org/abs/2310.17514 they are all speaking different languages and talking past each other.
>
> W1: We will make this clearer.
>
> W2: We will run this additional experiment. As you note, we expect to see very similar results.
>
> W3: We will add a clearer explanation of gumbel-softmax and its implications.
>
> W4: We will move the interpretability section to the main body and elaborate on it in more detail.
>
> W5: This uses standard categorical representations and cross-entropy loss.

---

> > ### Comment · Reviewer_U78F · 2025-02-04
> >
> > Integration: Please do include these, at least in the appendix! This is crucial for understanding the potential impact of the work
> >
> > Length Generalization: I would need details describing how this module can be integrated into such architectures in order to appreciate this point. As I see it, the most obvious thing to do would be directly replacing the FNNs of a transformer with SMFRs. This would require the blocks to operate within individual tokens, and it isn't obvious that this is useful for compositional processing at the level of sequences (which is the level of granularity that compositional semantics is most often discussed).
> >
> > Benchmarks: This reference supports the idea that four common compositionality datasets do not necessarily agree with one another, but does not suggest that one should give up on all existing datasets. The authors seem to know exactly what sorts of compositional capabilities they are capturing, and so should be able to make predictions about whether their model should succeed or fail on particular compositionality benchmarks, and why. For example, the systematicity split of PCFG SET from [1] seems to reflect aspects of the algorithmic experiments from this work, and so we might expect success at this subtask. Including some existing benchmarks and clear explanations for SMFRs success or failure on them would make for a much stronger contribution.
> >
> > W5: When is this applied: After each of the two iterations? Or only after both iterations? These details are opaque in the main body, and should be clarified.
> >
> > [1] Hupkes, Dieuwke, et al. "Compositionality decomposed: How do neural networks generalise?." Journal of Artificial Intelligence Research 67 (2020): 757-795.

---

> > > ### Author Response · Authors · 2025-02-04
> > >
> > > Thank you for your fast response!
> > >
> > > Length Generalization: Your interpretation is roughly correct. What you describe will fix the problem that transformers can only perform pairwise comparisons, which hampers their routing abilities. However, there will be an additional benefit: Block-operations will ensure that later layers can easily reuse the representations used by earlier layers. Two previous improvements to the transformer architecture addressed this exact point to a lesser degree and both improved performance: Transformers now contain a residual connection, and (Csordás et al., 2021 ; https://arxiv.org/abs/2108.12284 ) showed that using learned gated weights for the residual connections further increases performance (similar to how our FNNR functions).
> > >
> > > Benchmarks: We spent a lot of time choosing between benchmarks, but all of them have confounders or require additional architectural changes with their own hyperparameters. For example, the PCFG SET benchmark is a sequence prediction problem. To address this, we would have to adjust an RNN, LSTM, or Transformer to block-operations. This would have added even more complexity to an already complex architecture. We wanted to keep it simple for the initial publication, which was why we chose to create tests for fundamental properties instead.
> > >
> > > W5: The loss is only applied after every two iterations. This is why it is impressive that it also generalizes to odd-numbered iterations. We will make this clearer.

---

### Review · Reviewer_Hg1i · 2025-02-12

**Summary Of Contributions:**

The paper introduces a new architecture to process blocks of computations and improve compositional generalization. Each layer (MFNNR) is based on a Multiplexer and a Feedforward Network enhanced with gated residuals (FNNR), forming the final architecture.
The claimed properties and performance of the model have been analyzed using synthetic and realistic scenarios and compared to some baselines such as a feedforward NN.

**Audience:**

Yes

**Broader Impact Concerns:**

No concerns.

**Claims And Evidence:**

No

**Requested Changes:**

## Architecture Motivation and Explanation of the Mechanics
I understand that Figure 1 is a bit of a sketch to exemplify the possible output blocks of SMFR, but I suggest the authors expand it and avoid relying on color coding. At the current stage, I don’t find it particularly useful as a “teaser.”

The basic operations used as motivation (i.e., copy, FNN output, and linear interpolation) are interesting, but it would be better if experiments supported them. Am I understanding correctly that the point here is that the proposed architecture should, in principle, be able to perform these operations while the classic FNN cannot? But what about transformers?

Figure 2 could also be improved. It is unclear what the input blocks of the FNNR are (maybe the output blocks of the multiplexer?), and what about the extra_inputs? From the text, it seems these are the input blocks of the multiplexer. Is this always the case?

I honestly found it difficult to map the proposed architecture to known concepts such as self-attention and multi-head attention. To me, the multiplexer is computing something similar to N global attentions across input blocks, where the coefficients are generated using an FNN applied to the input blocks. Please correct me if I wrongly interpreted the model.
In general, I think the paper would benefit from a one-to-one comparison with the components of the transformer block to explain why the proposed architecture is needed and when it is preferable to the transformer and the self-attention operation.

Another way to improve the paper is to use formulas and pseudo-code to describe the operations involved in the proposed blocks. For instance, it is unclear (at least to me) in which dimension (axes) the blocks are created and whether the FNN is applied to the concatenation of the input blocks. All these details should be clear from reading the paper.

## Experiments
**Section 5.1**
I acknowledge that evaluating compositional generalization, and more specifically, the properties of a new modular architecture, is challenging. Nevertheless, I believe the experiments could be presented in a better way. First of all, Table 1 (and also Table 2, it seems) is an aggregated table across different architectures and model sizes, which is a bit odd because model size could be considered a hyperparameter, and it is necessary to analyze its impact, especially for a new architecture. In my option, Table 4 should be included in the main paper and expanded with the aggregated metrics, rather than being relegated to the appendix.

- It is unclear how the model size (number of parameters) relates to the hyperparameters of the model (number of layers, width, number of blocks, etc..). This is a point that should be improved. In this regard, in Appendix A.2.1, this sentence is unclear and should be expanded as it is somewhat vague: _“It should be noted that the model sizes of FNNs and SMFRs were varied in different ways. The table indicates that growing SMFRs by stacking more MFNNR modules together is not as effective as increasing the FNNs inside the SMFRs.”_

- This is more of a general remark, but it applies to Tables 1 and 4: it would be important to evaluate the convergence speed of SMFR vs. FNN vs. Transformer. In this particular experiment, my question is mainly about understanding how the number of training iterations affects negative interference. As you fix an accuracy threshold, one architecture may be faster than another (more sample efficient); hence, the number of iterations and data could affect the results.  Why is the transformer not tested in these experiments? It would be important to test other architectures for compositional generalization.

**Section 5.2**
I suggest the authors expand this section with at least a table of results. Leaving all results in the appendix makes the paper difficult to navigate. When mentioning a specific additional result, I suggest referring to the specific section of the appendix.
Why are the transformer and other architectures not tested in these experiments?

## Interpretability of the Proposed Architecture
I think this section should be moved to the main paper and expanded with plots and visualizations to make it more impactful. These are the kinds of analyses that should be performed when designing a new architecture. Interpreting the model and understanding its mechanics are important to motivate the adoption of a new architecture over the state of the art. For instance, similar analyses could be extended to examine the failure cases of other architectures, such as the transformer.

## General Remarks
A general remark is that the paper is missing visualizations that could make it much more enjoyable to read and easier to digest. For instance, Table 3 would be much more interesting if integrated with a plot.
I also suggest discussing the effect of varying the hyperparameters for each experiment. The appendix only partially covers this, but it is difficult to relate the effect of the hyperparameters on model performance (e.g., the missing accuracy from Table 6).
As stated above, it would be great to have plots about convergence speed, for instance, comparing loss functions and accuracies for different hyperparameters and models.

**Strengths And Weaknesses:**

**strengths**
- interesting architecture to improve compositional generalization
- different experimental settings in a controlled way (but presentation could be improved)
- The interpretability section adds value by analyzing model behavior, which is crucial when proposing new architectures, although it should be expanded to motivate to support the design choices of the architecture.

**weaknesses**
- The paper's presentation, in particular when discussing the model and the organization of the results could be improved,
- Confusing experiment details, insights, and almost no visualization.
- Possibly missing baselines: transformers and other architectures for compositional generalization.
- It is unclear, if and why transformers are not able to generalize as the proposed architecture and if this is fully supported by the experiments.

---

> ### Author Response · Authors · 2025-02-13
>
> Dear Reviewer,
> Thank you for your constructive feedback.
> - Regarding transformers, baselines, and experiments:
> We mention in related work: “Attention in Transformers is based on pairwise comparisons. This makes it difficult for them to learn routing mechanisms based on more than two features, such as conditional statements.”
> This is confirmed by our experiments, where transformers fail exactly for the tasks where pairwise comparisons are insufficient.
> - Regarding comparisons of convergence speed:
> This is an excellent point! We will run these experiments again and report the convergence speeds. Thank you for pointing out this gap in the experimental procedure. We will also test the transformer on these experiments for completeness.
> - Regarding presentation:
> Many of the presentation issues you highlight overlap with criticism raised by other reviewers. We will be happy to improve our presentation and to add clarifying details where they were requested. We will also move text from the appendix to the main body and add additional details as requested.

---

> > ### Comment · Reviewer_Hg1i · 2025-03-03
> > **new revision?**
> >
> > Dear Authors, thank you for your reply.
> >
> > As the deadline for the final recommendation is approaching, should we expect a revision of your manuscript with the requested changes?

---

> > > ### Author Response · Authors · 2025-03-04
> > >
> > > Dear Reviewer,
> > >
> > > Due to capacity constraints we will unfortunately not be able to make the requested changes in time.

---

### Decision · Action_Editor_fYpR · 2025-03-19

**Recommendation:** Reject

**Comment:**

All reviewers of this paper recommended rejection for the reasons outlined under "claims and evidence".

**Audience:**

Reviewers did think it was possible that it could be of interest to some of TMLR's audience.

**Claims And Evidence:**

The reviewers did not feel that the paper provided sufficient evidence to support the claims. Specifically, reviewers were concerned about insufficient comparison (in terms of both motivation as well as empirical results) to existing architectures as well as unconvincing experimental results (focusing mainly on toy problems rather than established benchmarks that measured compositional generalization). While reviewers could had some understanding of the benefits of the proposed architecture, they also found the architecture's description to be unclear and also addressed significant limitations compared to existing architectures.